# Possible Mental Health Interventions for Family Members of a Close Relative Who Has Suicidal Thoughts or Committed Suicide: A Pilot Project at a Mental Health Center

**DOI:** 10.3390/jcm13072032

**Published:** 2024-03-31

**Authors:** Sigita Lesinskienė, Virginija Karalienė, Kamilė Pociūtė, Rokas Šambaras

**Affiliations:** Clinic of Psychiatry, Institute of Clinical Medicine, Faculty of Medicine, Vilnius University, 01513 Vilnius, Lithuania; virginija.karaliene@gmail.com (V.K.); kamile.pociute@mf.vu.lt (K.P.); rokas.sambaras@mf.vu.lt (R.Š.)

**Keywords:** suicide, suicide attempt, self-harm, family members, mental health care, program

## Abstract

**Background:** Suicides are an actual issue, especially in Lithuania, where, despite significant efforts, the number of suicides remains very high. In cases of suicide, society painfully loses its members, and the relatives of the person who committed suicide, engaged in self-harm, or attempted suicide face many negative experiences. **Methods:** The purpose of this article is to describe the pilot project applied for 2 years in the Mental Health Center (MHC) in the city of Vilnius, Lithuania, in organizing and providing services to people whose relatives committed suicide or attempted suicide or self-harm. This prevention and early intervention program, organized by an interdisciplinary team at an MHC, appeared appropriate, effective, and well-attended. Program clients could participate anonymously and have the opportunity to choose the time and the right services for themselves from the offered program services package. **Results:** Providing the possibility for family members to receive flexible mental health services at the primary center increased the attractiveness of the program and reduced stigma. **Conclusions:** The program results demonstrated the applicability of the implementation of such an initiative as a relevant possibility when providing complex help for the relatives of self-harming and/or suicidal people.

## 1. Introduction

Suicide takes a staggering toll on public health, with almost 700,000 people dying from suicide worldwide every year. This figure may be significantly more remarkable due to the possibility of underreporting [1]. Suicide has become a prominent health and societal issue in many countries. Data from the World Health Organization (WHO) show that the global age-standardized suicide rate was 9 per 100,000 population in 2019, with rates varying between countries from fewer than 2 deaths by suicide to over 80 per 100,000 population [1]. Suicide rates in Africa (11.2 per 100,000), Europe (10.5 per 100,000), and Southeast Asia (10.2 per 100,000) were higher than the global average in 2019. The lowest suicide rate was in the Eastern Mediterranean region (6.4 per 100,000) [1]. In addition, it is important to emphasize the crisis of suicide among young people. Suicide is the second leading cause, and in some European countries, the leading cause of death amongst young people aged 15–24 years, exceeding the number of accidental deaths [2]. There are indications that for every adult who dies by suicide, there may be more than 20 others who attempt suicide [1]. Therefore, it is essential to talk about not only people who have committed suicide but also those who are trying to commit suicide or thinking about suicide. It is also necessary to mention self-injurious behavior because research findings highlight that non-suicidal self-injurious behavior may be predictive of suicidal ideation and behavior [3]. Suicidality (wishing to be dead, having suicidal ideation, planning suicide, attempting suicide, engaging in non-suicidal self-injury) is a severe public health and social problem that unfortunately does not seem to be diminishing [4].

Research has demonstrated a need for increased awareness and knowledge about suicide and suicide prevention among politicians in five European countries [5]. A positive result was that the state’s enhanced understanding of the importance of suicide prevention created the conditions to initiate and fund suicide prevention efforts in our country.

Lithuania is one of the leading countries in the world in terms of the number of suicides. According to Eurostat data, the age-standardized suicide rate in Lithuania was 21.25 per 100,000 population in 2020, which was more than twice the European Union average (10.24 per 100,000) [6]. However, the number of suicides in Lithuania has been decreasing in recent years. Compared to 2004, when the age-standardized suicide rate was 42.89 per 100,000 population, the suicide rate was lower by half in 2022. Based on the most recent data available from the Lithuanian Institute of Hygiene, the age-standardized suicide rate was 18.6 per 100,000 population in 2022; that year, 527 people committed suicide in Lithuania (426 men and 101 women) [7]. Over the past decade, suicides of both men and women have decreased in Lithuania. However, there were many more suicides among men: in 2012, the suicide rate for men exceeded the rate for women by 5.1 times, but in 2021, it decreased to 3.9 times [8]. However, the WHO indicates that the average suicide rate for men and women in high-income countries is close to three [1]. Also, over the past decade, suicides have decreased in all age groups, except among those 70 years of age and older; the suicide rate for this age group has remained steady. Suicide rates decreased the most (by almost half) among children and adolescents (up to 19 years old) and middle-aged people (35–49 years old). The rate of suicide among young adults (ages 20–34) decreased by a similar amount, almost by half [8]. Even though the suicide rate for children and adolescents has decreased, it remains a critical health problem in Lithuanian society. Based on data from a study carried out in Lithuania, suicide accounts for a quarter of all deaths among adolescents aged 10–19 years [9]. 

To reduce the suicide rate in Lithuania, the government and non-governmental organizations have taken responsible actions. In 2014, the Lithuanian government published the Lithuanian Health Strategy for 2014–2025. This strategy was aimed at reducing the standardized suicide rate to 19.5 per 100,000 population in 2020 and to 12.0 in 2025. In 2015, the State Mental Health Center (MHC) established a structural unit, the Suicide Prevention Office. The function of this unit is to analyze and summarize information about suicide and risk factors, provide ongoing suicide prevention and postvention projects and programs, and conduct epidemiological studies of suicide. Municipalities were tasked with creating annual plans for their suicide prevention strategy. The focus was on suicide prevention training programs called safeTALK and Applied Suicide Intervention Skills Training (ASIST), which help “gatekeepers” learn to recognize signs of suicide risk and provide individuals with knowledge of where and how to seek help. The Collaborative Assessment and Management of Suicidality (CAMS) manual was translated into Lithuanian and published in 2017, and an effective CAMS training program was successfully organized. CAMS is a model of special support for individuals with suicidal ideation that describes how to assess the risk of suicide and create and apply a support plan to prevent suicide effectively and collaboratively. 

In 2018, the Lithuanian Ministry of Health enacted a resolution on the procedure for assisting persons at risk of suicide. The resolution provides for a clear plan of action for anyone at risk of suicide. All supporting institutions (ambulance services, primary health care centers, mental health care centers, tertiary hospitals) are included in this plan. The plan consists of action steps for each institution and the procedure for organizing assistance. One of the most critical parts of the plan is a psychosocial assessment of the patient or person who is considering suicide, which is carried out by a specialist, such as a psychiatrist, child and adolescent psychiatrist, or medical psychologist. To complete a psychosocial assessment, the specialist uses the Psychosocial Assessment of Suicidality Risks form or the CAMS Suicide Status Form. If the specialist determines that there is a high risk of suicide, the patient can be guided to inpatient treatment, and if the risk is low or medium, the specialist can access a safety plan. A safety plan provides an excellent visual reminder of the warning signs of suicide, coping strategies, support systems, professional agencies, family support, and emergency services. Persons with a safety plan are directed to the MHC according to their residence. The case manager of the MHC is required to contact anyone who is at risk of suicide within three working days from the date of the psychosocial assessment. Anyone who is at risk of suicide can receive at least 10 medical psychologist consultations at the MHC. The first consultation must be given no later than 14 days after the patient’s registration.

In Lithuania, MHCs are established in municipalities on a territorial basis and are staffed by primary teams of specialists: adult psychiatrists, child and adolescent psychiatrists, psychologists, social workers, and nurses. Some MHCs also employ broader interdisciplinary teams of specialists and have art, music, dance–movement, and drama therapists incorporated into the team. A network of MHCs throughout the country provides services for adults and children/adolescents and is a reasonable basis for implementing specialized programs and initiatives. 

A total of 111 MHCs are operating in Lithuania, and their activities are organized in accordance with the guidelines and requirements set by the Ministry of Health [10]. MHC activities are covered by the Compulsory Health Insurance Fund budget. For a small country with relatively limited resources like Lithuania, the possibilities include strengthening the existing network of services and finding opportunities for mixed financing models, which still need to be sufficiently implemented. There is a lack of stable, mixed, long-term funding for mental health prevention and intervention programs.

Actively implementing existing policy regulations and assuring service quality remain the main directions for further development of the services provided by MHCs. Funding for complex service provision and specialist training (team building and competencies) could be increased using more flexible financing by municipalities and governments. Help for more targeted patient groups could also be provided more flexibly. Responding to this need, one of the initiatives of MHCs, with the support of some additional funding from municipalities, was to organize an assistance program for family members whose loved ones committed or attempted suicide or self-harm, including both prevention and early intervention aspects.

The issues surrounding mental health are complex and often stigmatized, and they involve not only health aspects but also education and the social environment. A focus on destigmatizing MHC services is also growing in the country among media professionals, public opinion makers, politicians, administrators, and specialists. Sharing good examples from the MHC work organization would encourage the development of good initiatives and expand the options for responding to the needs of customers and patients. 

Despite the established suicide prevention system in Lithuania and interdisciplinary cooperation, various difficulties still remain. Mental health services are paid for by the health insurance fund, which brings a lot of bureaucracy into their provision and organization. Also, it has been noted that there is a particular lack of preventive programs aimed at persons whose relatives are facing a suicidal crisis. Family members themselves often do not seek help because they experience much stress and emotional tension when going through the grieving process, with complicated interpersonal relationships and emotional confusion in the situation where someone close engages in self-harm, wants to commit suicide, or has committed suicide. Interventions for family members could provide additional support that could help reduce the risk of suicide [11]. When services are financed by separate municipal programs, service providers, especially MHCs, can flexibly organize interventions and enable patients or their family members to get involved in ongoing programs.

The topics of mental health, self-harm, suicidality, and mental disorders are sensitive and delicate in families and society. Stigma is one of the main reasons why people do not seek help for mental health problems, especially when they experience someone close attempting suicide. The unique network of primary mental health outpatient centers in Lithuania, which provide services for adults and children/adolescents, is a reasonable basis for implementing specialized preventative programs and initiatives in this field [10]. The primary location of MHCs in general outpatient clinics (polyclinics) helps reduce isolation and stigma and the perception of a lack of psychosocial support.

The aim of our study was to describe and analyze a pilot suicide prevention program applied for 2 years at an MHC in the city of Vilnius, Lithuania, and compare it to other science-based prevention strategies. 

## 2. Materials and Methods

### 2.1. Program Description 

This pilot program was conducted in Vilnius, the capital of Lithuania. According to the Lithuanian Statistics Department, Vilnius has a population of 581,475, and it is the largest city in the country of Lithuania. This pilot program was part of the Vilnius City Suicide Prevention Strategy 2020–2024 plan. One of the MHCs in the city of Vilnius was chosen for its implementation. The main goal of this pilot program was to target individuals whose family members had suicidal thoughts, self-harmed, or committed suicide. The institution’s administration was permitted to conduct this study and to analyze and summarize the results. The goal was to prevent mental health complications through various health-promoting interventions that were accessible to Vilnius city residents, both children and adults. The project consisted of two parts: (1) services provided at the MHC and (2) lectures in schools on the topic “Me and My Emotions: How Can I Help Myself” (Figure 1).

### 2.2. Services Provided at Mental Health Center

The program’s participants included children (7–17 years old) and adults (18–69 years old) who had a close relative who had suicidal thoughts, engaged in self-harm, or committed suicide.

Before starting the program, participants (adults and children with parents) completed a questionnaire indicating their age, current emotional state, and preferred support methods. After the program, participants completed feedback questionnaires (Did you benefit from the project? Would you like to participate (would you like your child to participate) in this type of project again? How did you feel at the beginning vs. the end of the project? How did your child feel at the beginning vs. the end of the project?). The survey questions were designed to evaluate the well-being of the participants, and the responses were ranked on a Likert scale (0 to 10, where 0 is very bad and 10 is very good). The participants had the opportunity to choose different types of support according to their difficulties and needs. The program in 2023 provided individual counseling (psychiatrist or child/adolescent psychiatrist, medical psychologist, social worker) and group sessions (psychoeducation/supportive group to improve emotional well-being, water drawing therapy, group psychosocial interventions, kinesiotherapy, public health specialist group). Table 1 summarizes the different types of services offered in the project and the minimum number of consultations or group sessions that participants could choose.

The psychoeducation and emotional well-being support groups were led by a clinical psychologist and focused on learning about emotions, recognizing emotions, and managing stress. These groups were designed to create an atmosphere of mutual support, sharing difficulties and finding solutions together. Groups for children included more game activities, while groups for adults were more discussion-based. During the water drawing therapy, which was led by an art therapist, a different theme was presented each time, after which the participants would draw, followed by a collective discussion of the drawings. The psychosocial intervention group, which was led by a social worker, was aimed at improving social skills, having discussions, and developing skills to deal with everyday difficulties. Kinesiotherapy sessions were delivered by a kinesiotherapist in a gym at the polyclinic. The exercises were adapted to different people’s abilities, and education about the benefits of sport was also included. The public health sessions, which were delivered by a public health specialist, focused on education and motivation and skill building for healthier living, i.e., healthy eating, good sleeping habits, exercise, and other emotional health care. All groups lasted about an hour. During the consultation with the social worker, psychologist, or psychiatrist, the participant discussed their difficulties, and help was provided according to their needs.

### 2.3. Lecture in Schools: “Me and My Emotions: How I Can Help Myself”

Six schools in Vilnius were involved in this part of the program. The lecture was tailored to children up to 12 years old and adolescents aged 13–18 years. These lectures on emotions and stress management were an essential part of the students’ education, imparting crucial skills and strategies that the students could use in their daily lives. After the lecture, children and adolescents filled out a feedback form indicating what they liked most, what new knowledge they gained, and where they could use it.

### 2.4. Statistical Analysis

The statistical analysis (comparisons) was performed according to the background variables. Quantitative variables were expressed as mean values, and qualitative data were reported as numbers and percentages. The paired samples t-test was used to assess the significance of changes in emotional state (beginning vs. end of the project). Differences were considered statistically significant when *p* < 0.05. 

IBM SPSS Statistics (version 22) and Microsoft Excel 2016 were used to analyze the data.

## 3. Results

### 3.1. Services Provided at the Mental Health Center

In total, 42 children (31 girls and 11 boys) and 37 adults (33 women and 4 men) participated in 2023. The program provided 158 individual consultations and 210 group interventions. The number and characteristics of interventions are summarized in Table 2. 

Adults were more likely to choose the supportive group and the psychosocial intervention group and less likely to choose kinesiotherapy. Children were more likely to choose the psychoeducation and psychosocial intervention groups and less likely to choose kinesiotherapy. Children were more likely than adults to choose water drawing therapy, and adults were more likely than children to choose a public health specialist.

Most respondents found the project helpful and indicated that they would participate again. There was a statistically significant difference in means (*p* < 0.05) when comparing children’s and adults’ well-being before and after the project (Table 3).

Participants rated the project activities positively, noting that they were encouraged to get involved because their disorders would not be recorded, they would be free to come and talk, and the services were free of charge. The evaluation of the feedback questionnaire showed that specialist support was impactful for different age groups, with many participants stressing the importance of continuity. Participants valued a safe, supportive environment, the creation of trust, the absence of bureaucratic administration of health services, the avoidance of stigma, the opportunity to interact with professionals, and the ability to choose the most appropriate options for themselves from various interventions.

### 3.2. Lecture in Schools: “Me and My Emotions: How I Can Help Myself”

A total of 333 children attended: 208 aged 7–12 (97 boys, 93 girls, 18 refused to say) and 125 adolescents (63 girls, 51 boys, 11 refused to say). Among them, 88.9% of children and 90.4% of adolescents found the lecture useful, and 77.9% of children and 81.6% of adolescents said they would like to attend this type of lecture again. Both children and adolescents expressed that they would use the information they received from the lecture to help themselves and their friends and family members.

The majority of respondents liked learning how to recognize stress and how to deal with emotional outbursts. The most relevant topics were found to be psychological health, fear, anxiety, anger management, self-esteem, and relationships with parents and friends. Some respondents noted that they turned to friends/classmates for help/comfort when experiencing unpleasant emotions, but a significant number of them said they did not go anywhere when they needed help or turned to themselves. Many respondents said they experienced perfectionism, but this is not a significant difficulty or concern for them.

## 4. Discussion

This pilot project in Vilnius (Lithuania) perfectly fits into the recommended prioritization of initiatives for suicide prevention. According to the consensus position paper, European countries should initiate or reinforce national suicide prevention programs and task forces to decrease the social burden and significant economic impact of suicide [12]. According to the European position paper, the suicide prevention strategy should be separate from the general mental health strategy and have independent funding. The program has demonstrated beneficial results. Further implementation of the program could occur by maintaining the developed model and assuming that it is possible to shape it with flexible options to meet the community’s changing needs. 

For the participants of the program, there was the possibility to talk about and discuss their experiences with different services when facing the suicide attempt of someone close to them. Research demonstrates that exposure to the suicide of a close contact is associated with several adverse health and social outcomes, including an increased risk of suicide among partners bereaved by suicide, required admission to psychiatric care for parents bereaved by the suicide of an offspring, suicide of mothers bereaved by an adult child’s suicide, and depression in offspring bereaved by the suicide of a parent [13].

The different choices of activities for children and adults shows the need to adapt mental health programs to the specific needs of the individual. The selection of kinesiotherapy by few participants raises questions. Scientific research indicates that there are great benefits of physical activity for mental health as well as suicide ideation, so it would be important to find ways to attract people to participate more in physical activities in such programs [14]. The fact that this prevention program was free improved access to help, especially given the literature findings on socioeconomic inequalities surrounding suicide in Europe [15].

Our program stands out in terms of both the quantity of services and the fact that interventions were tailored to a specific group with a possibly higher risk of suicide than the general population within the primary care setting: individuals with family members who had suicidal thoughts, engaged in self-harm, or committed suicide. It is known that this group may have a higher risk of suicide; for example, it was found that individuals with a relative who has attempted suicide have more suicidal thoughts [16]. It is difficult to find programs such as the one described in this paper that are implemented in a primary setting in the scientific literature. Most often, analyses of suicide interventions take place in primary care settings, such as providing specialist education, screening for suicide risk, treating depression symptoms, and assessing and managing suicide risk [17]. It is difficult to find a package of interventions in the scientific literature; more frequently, studies describe the effectiveness of separate interventions. For example, psychoeducation for suicidal patients is described as a promising intervention [18]. A systematic review indicates that art therapy increases the likelihood of taking action to prevent suicide and decreases suicidal risk and self-harming behaviors [19]. Increased physical activity may be a protective factor for suicidality [20]. School-based programs promoting emotional well-being and social skills show promising results for emotional health [21]. There is a dearth of studies evaluating a combination of similar interventions. A systematic review published in 2016 examining suicide reduction programs, such as public and physician education, media strategies, screening, restricted access to means of suicide, treatments, and internet support, did not find that some interventions were significantly more effective than others. A total of 164 programs were analyzed, and the studies included a variety of populations, such as the general population, people with psychiatric illnesses, different age groups, different countries, etc. There were no studies that looked specifically at populations of people with relatives who had suicidal ideation, self-harmed, or attempted suicide. The authors concluded that a combination of science-based interventions is recommended [22]. More research examining combinations of interventions is needed to better understand the effectiveness of these types of programs.

When we look at the research done on children who have lost a parent to suicide, we can find some studies showing the effectiveness of group therapy. A study published in 2017 that attempted to review group psychosocial interventions for children and adolescents bereaved by suicide found only two relevant studies [23]. They reviewed Pfeffer et al.’s study, in which children and their parents were given 10 group interventions, and the effectiveness of the program was compared to a control group. Children who received the intervention experienced more pronounced changes in anxiety and depressive symptoms [24]. The second study evaluated was by Daigle and Labelle. The research looked at 12 group interventions for children. Positive changes in the percentage of children participating were observed, although statistical tests were not performed due to the small sample size. In contrast to the Pfeffer et al. study, improvements in social adjustment and post-traumatic stress disorder symptoms were observed [25]. We found a paper that presents the long-term results of a family bereavement program (FBP) for children and adolescents who have lost a parent to illness, accident, homicide, or suicide. In the study, families were randomly allocated to participate in either the FBP or the control literature (CL) program. The FBP consisted of 12 group sessions for caregivers, children, and adolescents and two individual sessions. The CL program consisted of three books on adjustment to grief and a study guide for parents, children, and adolescents. Compared to the literature control group, participants in the FBP group reported fewer suicidal thoughts and/or suicide attempts at 6- and 15-year follow-ups [26]. This study did not indicate to what extent the interventions were effective for children who had specifically lost their parents due to suicide and not for other reasons. Overall, research suggests that various group therapies are effective in influencing children’s emotional well-being or suicidality, but further research is needed to better understand the effects of group and other interventions on specific populations of children.

Part of the program we described was a school-based intervention, i.e., psychoeducation. Some evidence suggests that school-based interventions to improve mental health are often effective [27]. The SEYLE study to reduce suicide is a good example. The interventions in SEYLE consisted of Question, Persuade, and Refer, a training module for teachers and other school staff, the Youth Aware of Mental Health Programme (YAM), a program for schoolchildren, and a screening by professionals to which at-risk students were referred. YAM showed effectiveness in reducing suicide attempts and severe suicidal ideation among school-aged adolescents as part of SEYLE [28]. Evidence on school-based programs’ effectiveness is not robust. For example, a systematic review and network meta-analysis of school-based interventions to prevent anxiety and depression in children and young people did not show that such interventions could be very effective [29]. School-based interventions were a small part of our study, and further research is needed to find out how school-based health programs can be effective in addressing suicide.

In the case of adults whose family members have self-harmed or committed suicide, there is also a paucity of research on the effectiveness of interventions targeted exclusively at this population. One study reviewing research on grief after the suicide of a relative found that most studies (five out of seven) focused on group interventions, one on a combination of individual and group interventions, and one on an individual intervention. The interventions varied from two to sixteen weeks, with most lasting around eight weeks. Bereavement groups have shown effectiveness in reducing the intensity of uncomplicated grief, while writing interventions have been effective in addressing suicide-specific aspects of grief. Additionally, cognitive–behavioral programs have proven helpful for individuals experiencing high levels of suicidal thoughts within the grieving population [30]. It is difficult to compare the program described in our study with others due to methodological differences. More research is needed in this area, specifically targeting people whose family members have had suicidal thoughts, self-harmed, or committed suicide.

Building on the program’s results, we turned to a broader approach to suicide prevention in Europe to contextualize our findings. Surveys in eight Eastern European countries indicated that the number of adverse childhood experiences was positively correlated with subsequent reports of health-harming behaviors [31]. Over half of the respondents in this survey reported at least one adverse childhood experience, and having one adverse childhood experience increases the probability of having other such experiences. When the content of the current preventive program includes family, school, relationships with peers, and nurturing experiences during the program activities, there are more opportunities to respond to the complex psychosocial development needs of vulnerable children and adolescents. According to the literature findings, prevention programs to augment family and child resilience can have lasting effects on suicidal risk [32]. Recent data demonstrated that there is an association between bullying and suicidal ideation among elementary school children across Europe [33]. In the further development of this program, more attention could be paid to topics such as bullying and related issues when communicating with children at both schools and MHCs.

Research data in a pediatric setting revealed that an especially vulnerable subgroup comprises children and adolescents who have repeatedly attempted suicide. It appears that social and psychological support after a suicide attempt in the pediatric setting in Lithuania still lacks efficacy, as minors who repeatedly attempt suicide suffer from not being released. Paying attention to similar psychosocial burdens and elaborating on the targeted follow-up care are needed [34]. One good example of the targeted care of adolescents who attempt suicide is the SAFETY program, a cognitive–behavioral family intervention designed to increase safety and reduce suicide attempt risk; a 12-week trial of the SAFETY program in the USA demonstrated promising results [35]. The SAFETY program is rooted in a social–ecological cognitive–behavioral model; treatment sessions include individual components for youth and parents, with different assigned therapists, and family components to practice skills identified as critical in the pathway for preventing repeat suicide attempts among youths. These ideas and components could be adapted when implementing similar programs in other countries. Gradually, Lithuania has prepared the ground for further development in the field of suicide prevention.

More specialized programs could later evolve from this pilot program to address the needs of children, adolescents, and adults, specifically comprising such aspects as grief, bereavement, and stigma. Grief is a predictor of long-term risk for suicidal ideation and suicide attempts by children and adolescents bereaved by parental loss [36], and the effects of a family bereavement program demonstrated promising results 6 and 15 years later during follow-up evaluation [26]. Children who have lost their parents are a particularly sensitive group who need supportive and preventative interventions. Research has demonstrated that the mode of parental death and the child’s age at the time are associated with long-term risk of suicide and hospitalization for specific psychiatric disorders for the child [37].

Our program included various initiatives addressing the complexity of multiple aspects, and it is possible to supplement and develop the program considering relevant emerging aspects. Alcohol and drug abuse are also important related topics, especially since Lithuania lacks adequate services and methods of assistance, especially for children and adolescents. The data show that the alcohol policy situation can impact the suicide mortality rates in a given country, considering the well-known link between alcohol use and death by suicide [38]. Staff members who are organizing and implementing prevention programs should consider these interrelated topics in public mental health. In our program, the option to have individual consultations for family members allowed the participants to explore these sensitive topics individually. On the other hand, group activities turned out to be more suitable for teenagers.

Further development of the program could increase its potential and provide a good opportunity to crystallize more focused programs for targeted subgroups. In our study, the small number of men raises questions, particularly when the number of suicides is significantly higher for men than women. Further programs would aim to include more men, making such programs more attractive to them, and they could be among the priorities for suicide prevention [39]. Also, research data show that cancer patients are at increased risk of suicide, and the needs in this broad field are not adequately addressed [28,40,41,42]. In addition, research findings have demonstrated that pets and animal-assisted therapy could be beneficial in suicide prevention; the bonds between humans and animals can provide a relaxing experience, decrease psychological pain, and increase the sense of purpose and engagement [43]. 

Stigma-reducing efforts make the prevention program more attractive and flexible. The findings from the recent international studies in Lithuania also demonstrate the possible benefits of using the Personal Stigma of Suicide Questionnaire (PSSQ) in terms of both its construct validity and its utility in understanding barriers to help-seeking among those experiencing suicidality [44,45]. Participants experienced public stigma, self-stigma, and label avoidance. Literature analyses reveal that the stigma of suicide shares similarities with stereotypes of mental illness but also has some significant differences. Attempt survivors may be subject to double stigma, which impedes recovery and access to care [46].

A qualitative study in the UK analyzed the failure of suicide prevention in primary care and discovered that general practitioners tend to lack confidence in recognizing and managing suicidal patients and report structural inadequacies in service provision [33]. In this study, relatives highlighted failures in detecting symptoms and behavioral changes and the inability of general practitioners to understand the needs of patients and their social context. A perceived overreliance on antidepressant treatment is a major source of criticism by family members. The conclusion of this study was that mental health and primary care services must find innovative and ethical ways to involve families in the decision-making process for patients at risk of suicide [47]. Family members may be the ones who can clearly see the deterioration of a family member’s condition and recognize the risk of suicide, so their participation in treatment and prevention programs is essential.

The young people of Lithuania are at particular risk of suicide; data reveal that a non-intact family structure and weak family functioning are significant predictors of suicidal ideation and suicide attempts among adolescents [48]. Organizing a preventative program for family members increased the possibility of providing psychosocial help and support without seeing them as patients but as clients.

Program providers must be well-trained to recognize and assess suicidal risk. According to the literature, teaching clinicians the fundamentals of the risk assessment interview, comprising strategies for assessing a person’s existential state, imminent warning signs, the lethality of planned suicide attempts, and protective factors delivered with an empathetic, collaborative approach, is useful, especially in cases where suicidality is not obviously apparent [49]. 

There exist good practices in Lithuania for training medical specialists about suicide intervention and prevention: the data show that suicide intervention/prevention training could be beneficial for service providers [50], including emergency medical services [51]. Recent research shows that pharmacological interventions (ketamine, esketamine paroxetine, and buprenorphine) and non-pharmacological interventions (a crisis response plan and assertive case management) are beneficial for effective suicide prevention [52]. The systematic use of non-pharmacological interventions was introduced and strengthened in Lithuania. This program made it possible to expand the circle of aid recipients, including family members of people who attempted suicide, and this can be considered to have supportive, preventative, and early interventional aspects. Cost-effectiveness in the longitudinal period could also be considered an important issue for further investigation. 

Motivation and staff training were among the most essential issues during the program’s implementation. Periodic discussions and assessments of where we were during the process of preparing and implementing this program were helpful for team members. Tackling the sensitive and painful emotional problems of clients could bring the risk of secondary traumatization, depersonalization, and exhaustion for service providers. Burnout prevention should also be considered for service providers when planning to implement this program. Recent data demonstrate that burnout is a disturbingly and increasingly prevalent phenomenon in healthcare [53,54].

When designing support programs, it is important to know what sustains suicide attempts and use this information. A recent Australian study analyzed responses to open-ended survey questions about reasons for living among those with lived experience of suicide who were entering the suicide prevention workforce and found that connection to others and service were the most commonly stated reasons for living; other categories included being oriented toward future, life, self, pleasure, and spiritual values [55]. 

Research still demonstrates that there are concerns, imperfections, and irregularities related to suicide prevention and service organizations. In promoting research to support an integrated approach to suicide prevention and facilitate personal and population resilience, an international team of experts made 10 key recommendations, listed below [56]. These recommendations illustrate that the primary MHC in our project served as a relevant place and provided family members with the possibility of receiving help and supported suicide prevention. Better knowledge of the initiatives for developing relevant methods for suicide prevention could act as an impetus for further developments in the field. The early years of a child’s development and family and caregiver support are essential in fostering the resilience that protects individuals from suicide. Evidence-based treatment for the full range of mental disorders should be made available to the whole population to decrease suicide rates. There should be collaboration among schools, health services, social services, and law enforcement to provide health promotion and suicide prevention strategies for children and young people. Mental health literacy among the general public should be improved so that people are better equipped to recognize those in psychological distress and direct them to appropriate services, which would reduce stigma and ensure that people have access to early intervention. Primary care clinicians should be skilled and trained in how to screen for suicidal ideas in at-risk populations. Suicide hotspots should be routinely identified so that suicide prevention measures can be put in place and their effectiveness monitored. As people in general hospitals are at increased risk of suicide, all staff working in these settings should receive suicide prevention training as part of their mandatory training. People admitted to inpatient mental health facilities should have a routine, well-documented assessment of suicidality on admission and regularly thereafter and appropriate care plans put in place to better manage risk. As people in residential care homes are at increased risk of suicide, all staff working in these settings should receive suicide prevention training as part of their mandatory training. Individual cultural factors and spiritual beliefs, which can be a protective factor against suicide, should be identified and supported.

The program described in this paper covered many of the recommendations listed above. The program included young children (from the age of 7 years); it was non-specific, i.e., it did not target a specific disorder and was open to the general population; it included the services of a multidisciplinary team (psychiatrist, psychologist, social worker, art therapist, kinesiotherapist); schools were also involved in the program, educating children about emotions and stress management techniques; the program was also a targeted hotspot, as it included people at higher risk of suicide, i.e., those with a family history of suicide. The pilot program covered several aspects: prevention, emotional support, early intervention, education, and therapy. The flexibility to individually tailor the program to the client’s needs makes it more attractive and boosts attendance. This program did not include education from other specialists working in the field of mental health or other areas of the medical and non-medical sectors. 

Suggestions for future research may include collecting information from both children and adults on what interventions they find most acceptable when faced with the suicide of a family member. More research focusing on the specific group of people whose family members are suicidal or have committed suicide is important because there is a lack of studies conducted with this particular population. Further research could also include studies on the effectiveness of different combinations of interventions. Though longitudinal and randomized research is needed, our described program seems to address an existing gap in services for children, adolescents, and adults whose family members have had suicidal thoughts, engaged in self-harm, or committed suicide.

## 5. Conclusions

The program’s results demonstrate the benefits of implementing such an initiative as a relevant option when providing complex help for the relatives of people who engage in self-harm and/or commit suicide. In this program, it was important to create a trusting, safe, accepting relationship and environment and provide flexible service offerings according to the situation and needs of each client. Primary mental health centers could serve as good locations to help family members and organize preventative initiatives, reduce stigma, and be flexible models that can meet individual participants’ needs.

## Figures and Tables

**Figure 1 jcm-13-02032-f001:**
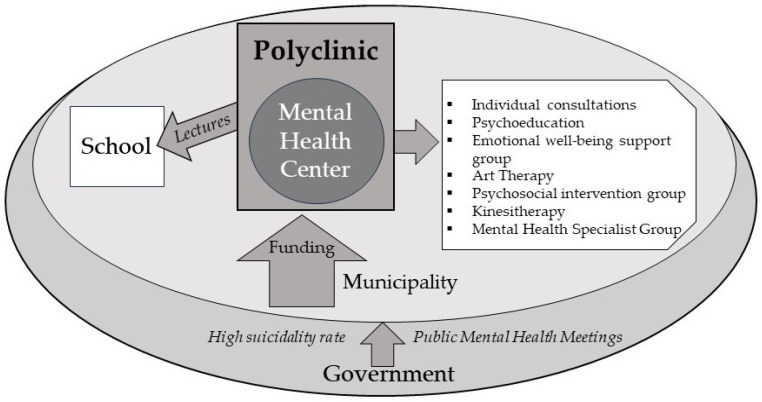
Scheme of pilot project.

**Table 1 jcm-13-02032-t001:** Minimum project requirements per person.

	Interventions for Child	Interventions for Adult
	Individual consultations	
Psychiatrist or child/adolescent psychiatrist or social worker	Minimum 1	Minimum 1
Medical psychologist	Minimum 1	Minimum 1
	Groups	
Psychoeducation	Minimum 6	-
Emotional well-being support group	-	Minimum 6
Water drawing therapy	Minimum 3	Minimum 3
Psychosocial intervention group	Minimum 6	Minimum 6
Kinesiotherapy	Minimum 6	Minimum 6
Mental health specialist group	Minimum 1	Minimum 1

**Table 2 jcm-13-02032-t002:** Provided services.

	Interventions for Children (n)	Interventions for Adults (n)
	Individual consultations	
Psychiatrist or child/adolescent psychiatrist	21	18
Social worker	21	19
Medical psychologist	42	37
	Groups	
Psychoeducation	36	-
Emotional well-being support group	-	36
Water drawing therapy	30	6
Psychosocial intervention group	21	15
Kinesiotherapy	23	37
Public health specialist group	-	6

**Table 3 jcm-13-02032-t003:** Feedback from participants.

	Participants (Children) (Mean of Scores)	Parents of Participating Children (Mean of Scores)
Did you benefit from the project? (0 = not at all, 10 = very useful)	8.2	7.9
Would you like to participate (would you like your child to participate) in this type of project again? (0 = not at all, 10 = very much)	8.3	8.8
How did you feel at the beginning vs. the end of the project? (0 = very bad, 10 = very good)	6.4 vs. 8 (*p* < 0.05)	5.8 vs. 7.9 (*p* < 0.05)
How did your child feel at the beginning vs. the end of the project? (0 = very bad, 10 = very good)	-	5.9 vs. 8 (*p* < 0.05)

## Data Availability

The data presented in this study are available on request from the corresponding author. The data are not publicly available due to privacy.

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
