# Peer review of "Possible Mental Health Interventions for Family Members of a Close Relative Who Has Suicidal Thoughts or Committed Suicide: A Pilot Project at a Mental Health Center"

_jcm, 2024, doi:10.3390/jcm13072032_

Round 1

Reviewer 1 Report

Comments and Suggestions for Authors

The paper, as described, offers a detailed account of a suicide prevention program in Lithuania, which is undoubtedly a relevant and pressing topic given the high suicide rates in the country. Nevertheless, based on the feedback, the originality of the paper seems questionable, with several opportunities for improvement highlighted.

Firstly, the paper recounts existing data and initiatives without offering significant new research or innovative methods. Although it is essential to build on previous work, the originality of a paper is greatly enhanced when it contributes novel insights or approaches. For instance, if the program introduced a unique intervention not seen in the literature or included a groundbreaking method of reducing stigma, such details would bolster the originality of the paper.

Secondly, while the subject matter is of importance, simply presenting it without novel insights limits the paper's contribution to the field. Originality could be elevated by proposing creative solutions to the challenges faced in suicide prevention in Lithuania, thereby setting a precedent that could be followed or adapted by others.

Thirdly, the lack of a clear statement of the program's uniqueness implies that it may not differ sufficiently from existing programs. Tangibly original elements might include a unique blend of psychological treatments, the application of new technologies or approaches in the interaction with relatives, or innovative community engagement strategies.

Fourthly, incorporating specific case studies from the program could lend the paper a degree of originality by providing rich, contextual examples of how the program works on a personal level. These narratives could illustrate its uniqueness and effectiveness in ways that aggregate data cannot.

Fifthly, a comparative analysis could have showcased the program's original aspects. This might involve contrasting the program's approach with those in different cultural or socioeconomic contexts or detailing how its structure addresses gaps that other programs have not bridged.

In sum, the originality of the paper seems limited by its reliance on describing existing approaches and a lack of emphasis on new methods or insights. To enhance originality, the authors could focus on distinctive elements of the program, perform rigorous comparative analysis, and integrate detailed case studies that demonstrate the impact of their work in a way that advances the field of suicide prevention.

Comments on the Quality of English Language

The feedback provided addresses distinct aspects of the language, grammar, and writing style in the article, with suggestions for improving clarity, flow, conciseness, and reader engagement. Based on the feedback, here is a critique focusing on those elements:

1. Clarity: The article appears to consist of verbose and complex sentences, which can obscure the meaning and make the text less accessible to readers. For instance, the sentence structure in the provided example involving participation statistics is convoluted. A clearer alternative might be: "In 2023, the program engaged 42 children (31 females and 11 males) and 37 adults (33 females and 4 males), delivering 158 individual consultations and 210 group interventions, as shown in Table 2."

2. Flow: The abrupt transition between topics hints at a lack of cohesion in the article. Efficient use of transitional phrases could mitigate this disjointedness. For example, a sentence like "Building on the program's results, we now turn to the broader approach to suicide prevention in Europe to contextualize our findings," serves as a bridge between two segments of the discussion.

3. Repetition and Redundancy: The article's repetitiveness in phraseology and redundancy in information can detract from reader engagement. Variety in language and a vigilant editing process to remove unnecessary duplication would benefit the article's impact and readability.

4. Conciseness: The critique suggests that some parts of the article are too descriptive, potentially overwhelming readers with detail. The challenge lies in balancing adequate context and succinctness. Refining the text to include only the most relevant details can enhance conciseness without sacrificing necessary information.

5. Grammatical Consistency: Inconsistent tense usage can confound readers and affects the professional quality of the paper. Consistent tense should be employed throughout to maintain coherence; for example, if the past tense is predominantly used to describe the conducted study, it should be used consistently.

6. Citations: Integrating references directly within the extract can bolster the article's academic integrity. By clearly citing sources where claims are made, the authors can add weight to their arguments and allow readers to verify information.

In summary, the paper's content seems to hold significance and presents valuable information on suicide prevention in Lithuania. However, there are opportunities to refine the language, enhance transitions between ideas, and ensure grammatical consistency to improve overall readability and effectiveness. The critique emphasizes the potential for a more polished and engaging article post-revision.

Author Response

Comments and Suggestions for Authors

The paper, as described, offers a detailed account of a suicide prevention program in Lithuania, which is undoubtedly a relevant and pressing topic given the high suicide rates in the country. Nevertheless, based on the feedback, the originality of the paper seems questionable, with several opportunities for improvement highlighted.

 Firstly, the paper recounts existing data and initiatives without offering significant new research or innovative methods. Although it is essential to build on previous work, the originality of a paper is greatly enhanced when it contributes novel insights or approaches. For instance, if the program introduced a unique intervention not seen in the literature or included a groundbreaking method of reducing stigma, such details would bolster the originality of the paper.

Thank you very much for noticing. The intervention and approach to reducing stigma in this program were that it was implemented and delivered in a primary mental health centre, and the organization of the program allowed participants to self-select appropriate activities from a suggested list in which they would like to participate. This was an additional possibility and novel activity, funded by the municipality, and participants of the program were treated not as patients but as clients. This is a good initiative and practice that the authors wanted to share in the manuscript cause the theme of suicides and organizing help for family members is of utmost importance. 

Secondly, while the subject matter is of importance, simply presenting it without novel insights limits the paper's contribution to the field. Originality could be elevated by proposing creative solutions to the challenges faced in suicide prevention in Lithuania, thereby setting a precedent that could be followed or adapted by others.

The original nature and success of this pilot program was that  ​​organizing help in primary mental health centers can be successfully developed and implemented more widely, both in Lithuania and other countries.

 Thirdly, the lack of a clear statement of the program's uniqueness implies that it may not differ sufficiently from existing programs. Tangibly original elements might include a unique blend of psychological treatments, the application of new technologies or approaches in the interaction with relatives, or innovative community engagement strategies.

An innovative community engagement strategy was applied by inviting family members to participate in the program. When they got acquainted with the possibilities of psychosocial help in the Mental Health Center, affiliated with Polyclinic, stiggme reducement and better understanding of the importance and possibilities of mental health services were provided. Such possibilities are very lacking in the medical field, where mental health services are funded and administrated from medical insurance money with very restricted and few possibilities for psychosocial help and support for family members. This gap could be improved by the flexible use of existing mental health team providers in Polyclinics. 

Fourthly, incorporating specific case studies from the program could lend the paper a degree of originality by providing rich, contextual examples of how the program works on a personal level. These narratives could illustrate its uniqueness and effectiveness in ways that aggregate data cannot.

Many thanks for the suggestion. Due to the large volume of the article and other aspects under consideration, we decided not to include case descriptions, but we plan to prepare another separate publication, presenting and discussing the clinical cases of this program.

Fifthly, a comparative analysis could have showcased the program's original aspects. This might involve contrasting the program's approach with those in different cultural or socioeconomic contexts or detailing how its structure addresses gaps that other programs have not bridged.

Authors have tried to catch and cover these aspects in the Discussion section of the manuscript.

In response to this comment, we have added an additional 7 articles and extra paragraph at the end of the discussion, demonstrating the originality, and scarcity of analogous programs. We outlined that our program encompasses a package of interventions rather than a single intervention, and that the service targets a specific demographic, which is underrepresented in the literature, thereby addressing gaps that remain unaddressed by other programs.

In sum, the originality of the paper seems limited by its reliance on describing existing approaches and a lack of emphasis on new methods or insights. To enhance originality, the authors could focus on distinctive elements of the program, perform rigorous comparative analysis, and integrate detailed case studies that demonstrate the impact of their work in a way that advances the field of suicide prevention.

Thanks for your observations. The authors tried to take this into account when adjusting the Methods, Results, and Discussion sections.

Comments on the Quality of English Language

The feedback provided addresses distinct aspects of the language, grammar, and writing style in the article, with suggestions for improving clarity, flow, conciseness, and reader engagement. Based on the feedback, here is a critique focusing on those elements:

  1. Clarity: The article appears to consist of verbose and complex sentences, which can obscure the meaning and make the text less accessible to readers. For instance, the sentence structure in the provided example involving participation statistics is convoluted. A clearer alternative might be: "In 2023, the program engaged 42 children (31 females and 11 males) and 37 adults (33 females and 4 males), delivering 158 individual consultations and 210 group interventions, as shown in Table 2."

  1. Flow: The abrupt transition between topics hints at a lack of cohesion in the article. Efficient use of transitional phrases could mitigate this disjointedness. For example, a sentence like "Building on the program's results, we now turn to the broader approach to suicide prevention in Europe to contextualize our findings," serves as a bridge between two segments of the discussion.

We have used a professional English language service to enhance the coherence of the text and strengthen its connectivity between sections. Additionally, we've incorporated your specific sentence into the discussion. Thank you for your valuable insights.

  1. Repetition and Redundancy: The article's repetitiveness in phraseology and redundancy in information can detract from reader engagement. Variety in language and a vigilant editing process to remove unnecessary duplication would benefit the article's impact and readability.
  2. Conciseness: The critique suggests that some parts of the article are too descriptive, potentially overwhelming readers with detail. The challenge lies in balancing adequate context and succinctness. Refining the text to include only the most relevant details can enhance conciseness without sacrificing necessary information.
  3. Grammatical Consistency: Inconsistent tense usage can confound readers and affects the professional quality of the paper. Consistent tense should be employed throughout to maintain coherence; for example, if the past tense is predominantly used to describe the conducted study, it should be used consistently.
  4. Citations: Integrating references directly within the extract can bolster the article's academic integrity. By clearly citing sources where claims are made, the authors can add weight to their arguments and allow readers to verify information.

In summary, the paper's content seems to hold significance and presents valuable information on suicide prevention in Lithuania. However, there are opportunities to refine the language, enhance transitions between ideas, and ensure grammatical consistency to improve overall readability and effectiveness. The critique emphasizes the potential for a more polished and engaging article post-revision.

According to your recommendation, the manuscript was edited by MDPI‘s English editing service.

On behalf of all authors, sincerely,

Prof. Sigita Lesinskiene

Reviewer 2 Report

Comments and Suggestions for Authors

Dear Authors,

The manuscript raises a very important and current problem of public health  not only in Lithuania. However, I suggest improving the "material and methods" chapter. In my opinion, sub-chapters 3.1., 3.2. and 3.3. they describe the material and method, not the results. The results are described in sub-chapters 3.4. (3.4.1., 3.4.2.).

Kind Regards

Author Response

Dear Authors,

The manuscript raises a very important and current problem of public health  not only in Lithuania. However, I suggest improving the "material and methods" chapter. In my opinion, sub-chapters 3.1., 3.2. and 3.3. they describe the material and method, not the results. The results are described in sub-chapters 3.4. (3.4.1., 3.4.2.).

Kind Regards

We want to thank you for your feedback. We expanded the Materials and Methods section: We have provided a detailed description of how the pilot program in MHC was organized. Also, we have organized the Materials and Methods section into subsections for better comprehension.

The manuscript was edited by MDPI‘s English editing service.

On behalf of all authors, sincerely,

Prof. Sigita Lesinskiene

Reviewer 3 Report

Comments and Suggestions for Authors

The only areas that I think could be more informative are the descriptions of the different programs/services that were included in the study. They were very briefly described .....yet given the focus of the paper, think how the reader would benefit from more focused and detailed descriptions of each service & then how they worked together for different populations. Such programs & services are critically important everywhere in the world.  I really think the paper has a great deal to offer - but the reader would get more from it with more descriptions.    

Author Response

Comments and Suggestions for Authors

The only areas that I think could be more informative are the descriptions of the different programs/services that were included in the study. They were very briefly described .....yet given the focus of the paper, think how the reader would benefit from more focused and detailed descriptions of each service & then how they worked together for different populations. Such programs & services are critically important everywhere in the world. I really think the paper has a great deal to offer - but the reader would get more from it with more descriptions.   

In response to this comment, we have included an additional paragraph in the methods section, providing a more detailed description of each intervention. Additionally, we have enhanced the clarity of the results. Furthermore, we have supplemented the discussion with another paragraph, which underscores the scientific efficacy of similar single interventions (similar to those described in our program) in various populations, as well as the significance and scarcity of combinations of such interventions for suicide prevention. Dear Authors,

The manuscript raises a very important and current problem of public health  not only in Lithuania. However, I suggest improving the "material and methods" chapter. In my opinion, sub-chapters 3.1., 3.2. and 3.3. they describe the material and method, not the results. The results are described in sub-chapters 3.4. (3.4.1., 3.4.2.).

Kind Regards

We want to thank you for your feedback. We expanded the Materials and Methods section: We have provided a detailed description of how the pilot program in MHC was organized. Also, we have organized the Materials and Methods section into subsections for better comprehension.

The manuscript was edited by MDPI‘s English editing service.

On behalf of all authors, sincerely,

Prof. Sigita Lesinskiene

Round 2

Reviewer 1 Report

Comments and Suggestions for Authors

The authors manage to ameliorate the manuscript, although the discussion needs further extension with future directions and description of relevant research. The recommendations provided at the end of the text need to be changed in a non-numerical order.

Author Response

The authors manage to ameliorate the manuscript, although the discussion needs further extension with future directions and description of relevant research. The recommendations provided at the end of the text need to be changed in a non-numerical order.

Many thanks for the revisions and valuable notes. We took into account all the comments and made the necessary additions and corrections: added seven references, including relevant articles for the discussion that describe relevant studies with children and adults who lost loved ones after suicide, and also school-based interventions. We added future directions at the end of the discussion and changed the text with statements in a non-numerical order. We also moved one part of the discussion text to another place of the discussion to make the text more fluent.
